# OpenReview forum: "Stepwise Alignment for Constrained Language Model Policy Optimization"
_NeurIPS.cc/2024/Conference — NeurIPS 2024 poster_

### Official Review · Reviewer_4XGF · 2024-06-19

**Soundness:** 2
**Presentation:** 3
**Contribution:** 3
**Rating:** 6
**Confidence:** 4

**Summary:**

The paper studies constrained policy optimization for the language model alignment problem. The authors propose a stepwise alignment method that involves two separate steps for fine-tuning a language model: first with a reward and second with constraints. Several advantages of the proposed method are illustrated compared to existing methods, such as simplicity, efficiency, and flexibility. In theory, the authors prove that the reward optimality gap and constraint violation are bounded, assuming linear reward/constraint functions. In the experiment, the authors demonstrate a practical implementation of the proposed method and show better performance than several existing algorithms.

**Strengths:**

- The authors characterize the optimization properties of the safe RLHF, such as strong duality. Although these properties are from the CMDP literature, they are particularly useful for understanding safe RLHF problems.

- The authors exploit the constrained RLHF problem structure to show that an optimal policy can be obtained in two steps. This is a useful property, as it allows us to improve existing models with safety constraints by using standard unconstrained RLHF algorithms (e.g., DPO, KTO).

- The authors also provide a theory of optimality for the proposed method. Although the linear function approximation assumption is restrictive, this appears to be the first theoretical characterization of safe RLHF.

- Despite the proposed algorithm being ideal, the authors provide a practical implementation and test its performance in several variations. Better practical performance is demonstrated through comparison with existing safe RLHF methods.

**Weaknesses:**

- The motivation relies on the existence of a safety model that can be constrained. However, in practice, how to set a constraint on the safety model is not discussed. This can be challenging since a safety model is often inaccurate, and the safety threshold is unknown.

- The multiplicative structure of the optimal policy in Theorem 1 assumes an optimal Lagrange multiplier. However, the analysis of the optimal Lagrange multiplier is not provided.

- The proposed stepwise method uses any Lagrange multiplier, raising a question: if an approximation of the optimal Lagrange multiplier is used, should we expect to achieve a similar near-optimal policy?

- The provided analysis implicitly assumes that an offline dataset can be represented by reward/safety models. However, it is challenging to verify the quality of offline data in practice. What happens if the models can't be represented by the data?

- The provided theory assumes linear reward and constraint functions, which can be restrictive in practice.

- The choice of an optimal Lagrange multiplier is heuristic in implementation, which is not characterized in theory.

**Questions:**

Below are some other questions for improvement.

- What is the meaning of *unparied* in line 110 and line 145? How are paired vs unpaired datasets determined in experiment?

- A mismatch in Eq (10): $\pi$ vs $\pi_\theta$. Do reward and safety models share the same parameter in line 120?

- What is the difference between Algorithm 1 and the multi-objective alignment $(r,g)$? Pre-selected $\lambda$ works as a preference.

- What other preference optimization algorithms can be used in Algorithm 1?

- What are some scenarios that involve multiple constraints? Can the authors provide experiments to illustrate these scenarios?

- How is the weighting scheme implemented for finding an optimal $\lambda$? What is convergence?

---

> ### Author Rebuttal · Authors · 2024-08-06
>
> We deeply appreciate the reviewer’s encouraging comments. We will answer the Questions first and then address the comments in Weaknesses.
>
> ### Questions
> **Unpaired vs paired.**
> In our paper, we call a dataset without paired preference labels {$y_w, y_l$} *unpaired*. Since this terminology may be confusing, we will add more explanations in the camera-ready version.
>
> **Eq. (10).** Thank you for pointing out our typo. As the reviewer mentions, a mismatch exists between $\pi$ and $\pi_\theta$. We will fix this in the camera-ready version.
>
> **Multi-objective alignment.**
> We consider SACPO to be advantageous for the following two reasons. First, multi-objective alignment with fixed $\lambda$ requires us to use the same algorithm (e.g., DPO) for both alignments on reward and safety. Our SACPO enables us to use different alignment algorithms for each metric (e.g., KTO for reward and DPO for safety). Second, multi-objective alignment needs a dataset that contains the set of outputs $\{y\}$ characterizing both reward and safety for each prompt {$x$}. Our SACPO is a stepwise approach, so the datasets do not have to share the same prompts. Such flexibility of algorithms and datasets is a major advantage of SACPO.
>
> **Other preference optimization algorithms.**
> SACPO is compatible with preference optimization algorithms developed for solving LM policy optimization problems with reverse KL penalty, namely Eq.(3). In this sense, $\Psi$PO, IPO (Azar+), or RSO (Liu+) can be directly applied as with DPO. Also, CPO (Xu+) or SimPO (Meng+) can be used in the first stage of SACPO. CPO or SimPO, representative reference-free algorithms, implicitly assume that the reference policy is uniform. The first stage of SACPO also does not require an explicit reference policy and is thus compatible with CPO or SimPO.
>
> - Azar+. "A general theoretical paradigm to understand learning from human preferences." In AISTATS, 2024.
> - Liu+. "Statistical Rejection Sampling Improves Preference Optimization." In ICLR, 2023.
> - Xu+. "Contrastive preference optimization: Pushing the boundaries of llm performance in machine translation." arXiv preprint arXiv:2401.08417 (2024).
> - Meng+. "Simpo: Simple preference optimization with a reference-free reward." arXiv preprint arXiv:2405.14734 (2024).
>
> **Multiple safety constraints.**
> Thank you for the great question! For example, we can consider a scenario where we want to obtain an LLM with less verbosity bias while making it generate helpful and harmless texts. We have conducted an additional experiment based on this scenario. **We have provided new experimental results in global responses** and hope they resolve the reviewer's concerns.
>
> **Weighting scheme for $\lambda$ and its convergence.**
> Weighting is implemented by the linear model merging of the reward-aligned model and conservatively safety-aligned model. In P-SACPO, the Lagrange multiplier $\lambda$ is optimized without necessitating additional training or fine-tuning of LLMs; thus, there is no notion of convergence.
>
> ### Weaknesses
>
> **How to set a constraint on the safety model.**
> One of the most likely scenarios is that the safety function $g$ is defined as a binary function ($1$ for safe prompt-generation pairs) and the threshold $b$ is defined as $0.95$ or $0.99$, for example. In this case, the safety constraint means that the probability of generating safe answers must exceed the threshold.
>
> **The analysis of the optimal Lagrange multiplier is not provided.**
> The multiplicative structure itself holds for an arbitrary $\lambda$. That is, the optimal policy under the reward function $r(x, y) + \lambda (g(x, y) - b)$ can be written as (11) where $r^\star$, $g^\star$ and $\lambda^\star$ are replaced with $r$, $g$ and $\lambda$, respectively.
>
> **Approximation of the optimal Lagrange multiplier.**
> Theorems 2 and 3 answer this question as long as Assumption 2 is satisfied. $\hat{\lambda}$ should be understood as an approximate value of the optimal $\lambda^\star$. If we only care about the discrepancy between $\hat{\lambda}$ and $\lambda^\star$ and ignoring the estimation errors for the reward and safety functions, $\hat{\Gamma}_{\mathcal{D}}$ in the theorem statements reduces to $|\lambda^\star - \hat{\lambda}|$. It implies that if $\hat{\lambda}$ is close to $\lambda^\star$, the reward of the obtained policy is close to that of the optimal one and the safety constraint violation is close to zero. In this sense, the obtained policy is close to the optimal one.
>
> **What happens if the models can't be represented by the data?**
> We explicitly stated in Assumption 2 that the true reward and safety functions are linear as well as the reward and safety models. In RLHF pipelines, the reward model is usually initialized from the SFT model by adding a linear layer on top of the final transformer layer to generate an estimated reward value. Thus, we do not consider that our theoretical analyses deviate from the actual scenario of LLM alignment. When such assumptions do not hold, LLM safety alignment will fail and then result in poor helpfulness or safety violations. At this moment we don't have any results for non-linear cases, and it would be a long-term research goal to extend our theory to more general function classes.
>
> **Linear assumptions.**
> It is true that assuming a linear reward model is restrictive. Since LLM alignment is a new research topic, our theoretical analysis is the first step toward the theoretical justification for SACPO. Relaxing this assumption is a remaining work in the future.
>
> **The choice of an optimal Lagrange multiplier.**
> As mentioned in Section 6, we do not consider optimizing both an LM policy and a Lagrange multiplier. This is our motivation behind P-SACPO, which enables us to find $\lambda$ without additional training or fine-tuning of LLMs. It would be a remaining yet interesting challenge to propose an algorithm that optimizes $\lambda$ in a theoretically grounded manner as an extension of the original SACPO.

---

> > ### Comment · Reviewer_4XGF · 2024-08-08
> >
> > Thank you for clarifying the choice of $\lambda^\star$. I have a follow-up question: when the order of your stepwise alignment is changed, how do you use the same averaging scheme? how to set $\bar{\lambda}$? The optimal policy may have different sensitivities to reward and safety objectives.

---

> > > ### Author Response · Authors · 2024-08-09
> > >
> > > We would like to express our sincere gratitude to Reviewer 4XGF who read through our responses.
> > >
> > > We consider that there are two approaches when the order of our stepwise alignment is reversed.
> > >
> > > The first approach is to regard the KL penalty coefficient for safety as $\beta$ and that for reward as $\beta/\bar{\lambda}$. In this context, our SACPO is effectively optimizing an LM policy with respect to $g + \bar{\lambda} \times r$. Hence, by setting $\bar{\lambda}$ a sufficiently large scalar, reward realignment will lead to a model emphasizing reward (i.e., helpfulness). In this setting, we can directly use P-SACPO as presented in Section 6.
> > >
> > > While we recommend the first approach, the second approach would be to implement P-SACPO using negation rather than addition. This idea is associated with a notion of task vectors, and negation leads to forgetting or unlearning (for more details, we would like the reviewer to see Figure 1 in [Ref]). Even when the order of the stepwise alignment is changed, we can use a similar $\bar{\lambda}$. Then we apply our stepwise alignment in the order from safety to reward. We now have two models with
> > > - Model A: Conservative safety
> > > - Model B: Conservative safety + Helpfulness.
> > >
> > > However, both two models are excessively conservative. Hence, we now apply the negation of Model A from Model B with some coefficient $q$; that is, Model B - $q \times$ (Model A - SFT model). By properly tuning $q$, we can obtain a model that balances helpfulness (i.e., reward) and safety.
> > >
> > > We appreciate the valuable questions raised by Reviewer 4XGF for improving the quality of our manuscript.
> > >
> > >
> > > [Ref] Ilharco, Gabriel, et al. "Editing models with task arithmetic." The Eleventh International Conference on Learning Representations.

---

> > > > ### Comment · Reviewer_4XGF · 2024-08-12
> > > >
> > > > Thank you for your clarification. As shown in Figure 2(b), different alignment orders lead to varying performance gaps, which can be quite large. It seems the authors do not have a complete answer for this. I expect that this issue will become more severe in the multi-constraint case, as different constraints introduce different sensitivities through the Lagrangian multipliers. I am wondering how challenging to set a large vector $\bar{\lambda}$ for multiple constraints. I am not sure how the stepwise method scales up to multiple constraints.

---

> > > > > ### Author Response · Authors · 2024-08-12
> > > > >
> > > > > We deeply appreciate Reviewer 4XGF for the insightful comments and constructive suggestions.
> > > > >
> > > > > As the reviewer mentions, different alignment orders lead to varying performance gaps. We hypothesize that the poor representation ability of the LLMs or optimization error regarding DPO might contribute to this phenomenon, though we do not have a definitive explanation. This represents an interesting direction for future research to analyze the gap between theory and practice. A similar comment was provided by VoMk, and we now concur with Reviewers 4XGF and VoMk that this is an important point. We will discuss this in the Limitation section of the camera-ready version.
> > > > >
> > > > > Regarding the questions about multiple safety functions, we currently do not have clear answers, and we sincerely acknowledge this as our honest response. Future work will be necessary to investigate the challenges of setting a large vector for multiple constraints or how the stepwise method scales up to multiple constraints.
> > > > >
> > > > > Finally, we would like to clarify that **our paper primarily focuses on a problem with a single safety constraint**. The theoretical results concerning multiple safety constraints presented in Appendix B are intended to facilitate further extensions of this work. Generally, extending an algorithm designed for a single constraint to one that handles multiple constraints is not straightforward. For instance, this complexity is evident in safe (or constrained) RL algorithms, where many algorithms inherently struggle to manage multiple safety constraints both theoretically and empirically. Based on the reviews, additional experiments, and discussion with reviewers, we now believe more strongly that our SACPO can serve as a solid foundation for future research due to the following reasons:
> > > > >
> > > > > - SACPO enjoys theoretical justification for the extension to multiple safety constraints.
> > > > > - We were able to conduct an additional experiment with two safety constraints within the short rebuttal period, demonstrating SACPO's stability and computational efficiency.
> > > > >
> > > > > We are grateful for your valuable suggestions and feedback, which have significantly contributed to improving the quality of our manuscript.

---

> > > > > > ### Comment · Reviewer_4XGF · 2024-08-12
> > > > > >
> > > > > > Thank you for your clarification. After considering your response and other reviews, I have updated my rating.

---

> > > > > > > ### Author Response · Authors · 2024-08-12
> > > > > > >
> > > > > > > We extend our heartfelt gratitude to Reviewer 4XGF for taking the time to consider our responses and other reviews carefully. Thank you for your constructive review and discussion!

---

### Official Review · Reviewer_w7hx · 2024-07-12

**Soundness:** 4
**Presentation:** 4
**Contribution:** 3
**Rating:** 7
**Confidence:** 4

**Summary:**

The paper presents the SACPO, a method that optimizes LM policies by sequentially aligning them to maximize helpfulness and harmlessness in either order. By selecting appropriate hyperparameters, the method enables balancing these criteria according to contextual needs. The authors leverage DPO and KTO in various experimental settings to demonstrate the effectiveness of their method, outperforming the prior SafeRLHF approach. SACPO is backed up with strong theoretical validations.

**Strengths:**

S1. SACPO is grounded in strong theoretical foundations, ensuring that the final policy is as effective as if it were optimized simultaneously for both objectives (Theorem 1). The use of the $\delta$-uncertainty quantifier to statistically bound errors in estimation adds a layer of reliability and predictability. By establishing that the error between the estimated and true functions is statistically bounded, SACPO provides a robust framework reliable within known limits. This is useful for safety guarantees of LM harmfulness.

S2. While the authors primarily focus on a single safety function to constrain LM harmfulness and do not empirically analyze scenarios with multiple safety constraints, they provide a theoretical framework that outlines how SACPO can be extended to accommodate multiple safety constraints. This facilitates further extensions of this work.

S3. Not explicitly mentioned by the authors, but after both optimization stages, if the necessity arises to further fine-tune the language model in either direction of helpfulness or harmfulness, then this option is available. For instance, if further data becomes available, the model is deployed in different contexts, or new requirements for helpfulness or harmfulness arise which can be incorporated into the prompt, then the optimization can continue in one of the directions to ensure that it remains effective and relevant.

S4. While the authors do not explicitly address further optimization after the initial stages, SACPO has the potential to facilitate further fine-tuning of the language model as needed. For example, if new data becomes available, the model is deployed in varying contexts, or evolving requirements for helpfulness or harmlessness emerge, the model can be further optimized.

S5. SACPO supports combining multiple algorithms for maximizing helpfulness and minimizing harmfulness. While they currently utilize only DPO and KTO, more powerful optimization techniques could be adopted as they are developed. This flexibility enhances the potential to further improve the effectiveness of SACPO.

S6. The flexibility granted by selecting an appropriate Lagrangian multiplier $\lambda$, KL penalty $\beta$, as well as the mixing ratio $q$ for P-SACPO enables effective balancing of helpfulness and harmlessness to meet different contextual needs and specific requirements. Fixing $\lambda$ eliminates the need for iterative adjustments and thereby adds stability by avoiding the oscillations and instability encountered with dynamically optimizing primal-dual methods.

**Weaknesses:**

W1. The extent of the evaluation is not clearly defined, particularly in terms of the number and variety of prompts used for testing. This makes it challenging to assess the generalizability of SACPO.

W2. I don’t think the evaluation of helpfulness and harmfulness should uniquely be separated to different prompts and responses. Surely, the prompts used in SafeRLHF [1] are specifically selected to ‘trigger’ harmful responses from LMs with higher likelihood, and this is important to detect. However, the goal, ultimately, is to have the model generate responses that are simultaneously helpful and harmless. Therefore, it makes more sense to evaluate these two criteria in parallel on the same prompts and responses. Otherwise, the policy might learn to simply detect prompts that contain ‘triggering’ clauses, and proceed to output very safe but less helpful answers. Contrarily, if the prompt appears safe, the policy can freely generate a maximally helpful answer, while having very low potential of being ranked as harmful.

W3. The description of the training protocol involving the PKU-SafeRLHF dataset in the optimization of the base SFT policy using DPO/KTO methods lacks clarity. Specifically, it is not clear whether the same data from this dataset is employed across both optimization stages for helpfulness and harmfulness. Additionally, the duration of the training process is not mentioned.

W4. Although used in SafeRLHF [1], I don’t think that solely relying on LLMs to evaluate helpfulness and harmfulness on question-answering problems is a reliable or objective method. Note, that SafeRLHF also incorporated other methods of evaluation.

W5. The method introduces several new hyperparameters that necessitate tuning or heuristic selection when trained for other contexts. The choice of optimization sequence, the KL penalty scaler $\beta$, the Lagrangian multiplier $\lambda$, and the mixing ratio $q$ complicate the setup.

[1] Dai, Josef, et al. "Safe rlhf: Safe reinforcement learning from human feedback." *arXiv preprint arXiv:2310.12773* (2023).

**Questions:**

Q1. What extra insight do the ELO scores provide? ELO would only be a meaningful metric if methods would be compared with each other, not if they are compared pair-wise with a single baseline.

Q2. The main results are presented using a win-rate metric in comparison to the SFT baseline, yet GPT-4 is employed to rate responses on a scale from 1-10. Why is the numerical scoring necessary if the primary evaluation criterion is simply determining whether the new policy outperforms the base one. Why not have GPT-4 directly select the better response from a pairwise comparison, rather than assigning individual scores?

Q3. Is the same set of data used for both stages of training?

Q4. How extensive and general is the evaluation?

**Limitations:**

The authors have briefly addressed the limitations of their work. I have pointed out further limitations in the weaknesses section of this review. My main concerns are W2 and W4.

---

> ### Author Rebuttal · Authors · 2024-08-06
>
> We deeply appreciate the reviewer’s encouraging comments. The thoughtful comments of Reviewer w7hx are valuable and help us improve the manuscript's quality. We will answer the Questions first and then address the comments in Weakness.
>
> **Q1 (ELO scores).** In Figure 3 - 5, when we compute the ELO scores, we do not only compare these models to the baseline SFT model, but we compare all models mutually.  The biggest reason why we present ELO scores in Figures 3 - 5 is to show the relative performance levels of our (P-)SACPO and baseline methods such as Safe RLHF (i.e., beaver-7b-v1.0) or DPO (H). As an important metric, we show the win rates against the SFT model in Figure 2. However, when the performance of the SFT model is poor, the win rates are sometimes insufficient for evaluating better-aligned models. Therefore, we present both win rates and ELO scores in our paper.
>
> **Q2 (Numerical scoring by GPT-4).** Thank you for your comments! As you mentioned, having GPT-4 directly select the better response from a pairwise comparison is also a valid choice. However, in this work, we choose to follow the evaluation protocol of Safe RLHF as much as we can for a fair comparison (based on their published source code). We also observed that such a choice does not significantly affect our evaluation result.
>
> **Q3 (Dataset).** Yes, we used the same dataset (i.e., PKU-SafeRLHF-30K) for both stages of training. Since this dataset contains multifaceted labels on helpfulness and safety, we used each label for each alignment stage on reward and safety. As the reviewer points out, this part was unclear; hence, we will add more explanations in the camera-ready version.
>
> **Q4 (Evaluation).** Our primary objective is to propose a simple, computationally efficient method as an alternative to Safe RLHF. Therefore, we conducted experiments to compare the performance of our SACPO and Safe RLHF as fair as possible. However, as the reviewer points out, our empirical supports are limited in terms of the base SFT model and datasets. Therefore, we conducted additional experiments using different base models and datasets. Regarding this, we conducted two experiments: one used LLaMA2 and hh-rlhf datasets, and the other used Pythia and the PKU-SafeRLHF-30K dataset. **For more details, we would like the reviewer to see the global responses** and we hope our new results resolve the reviewer's concerns.
>
> **W1 (Extent of the evaluation).** We used prompts from the AlpacaEval dataset to evaluate helpfulness and those from the Safe RLHF study to evaluate safety (i.e., harmlessness). To evaluate helpfulness, we used all 129 prompts from the 'helpful\_base' subset of the AlpacaEval dataset. To evaluate the safety, we used all 83 prompts used in the Safe RLHF study. In the camera-ready version, we will clarify the description of the evaluation process in Section 7.
>
> **W2 (Separated helpfulness and harmlessness).** Thank you for your insightful comments! We will explain why we used different prompts to evaluate helpfulness and harmlessness. First, when evaluations of helpfulness and harmfulness are coupled, safe models are likely to be evaluated as helpful. This means that safety-aligned models potentially obtain an unreasonably high evaluation regarding helpfulness. This is based on our observations in early experiments that DPO (S) were valued as more helpful than we humans thought. In real applications with AI systems based on LLM, most of the prompts are benign and it is also important to generate helpful answers for benign prompts. Therefore, we decided to use benign prompts from the AlpacaEval dataset to assess the helpfulness and red-teaming prompts from Safe RLHF studies to assess the harmlessness, considering that the quality of the prompts is preferable for each evaluation. As the reviewer mentions, it is a critical yet open problem how to evaluate safety-aligned LLMs. We will add more discussion in the camera-ready version.
>
> **W3 (Description of the training protocol).** We appreciate the reviewer's comments. Though we wrote down the experimental settings in Appendix G.3 and submitted a source code, more details should be provided in the paper to make it self-contained. Also, in our setting, the duration of the training process for each alignment step is about one hour. In total, it takes about two hours (i.e., for helpful and safety alignment) to obtain the final model. We will enrich such descriptions in the camera-ready version.
>
> **W4 (Relying on LLMs for evaluation).** Thank you for your insightful comments. Due to time and budget constraints, we could not conduct human evaluations. Though LLM-as-a-Judge is now a popular approach in typical alignment research, it is still an open problem whether the same conclusion can be obtained in the safety alignment research. We will add such a discussion in the camera-ready paper and leave human evaluations to future work.
>
> - Zheng, Lianmin, et al. "Judging LLM-as-a-judge with mt-bench and chatbot arena." Advances in Neural Information Processing Systems 36 (2024).
>
> **W5 (Additional parameters).** As the reviewer mentions, our SACPO necessitates additional parameters compared to safety-agnostic RL-free alignment methods such as DPO or KTO. However, we consider that parameters for SACPO are easier to tune than other safety-alignment methods. For example, in P-SACPO, $\bar{\lambda}$ can be set $\bar{\lambda} = 10$ as a typical choice. With common settings of $\beta = 0.1$ in standard DPO implementations, $\beta/\bar{\lambda}$ is set to be $0.01$. As for $q$ (i.e., the mixing ratio), we only have to execute linear model mergings without additional training or fine-tuning LLMs.

---

> > ### Comment · Reviewer_w7hx · 2024-08-08
> >
> > **Q1**. Thanks for the clarification. If the comparison is indeed done across all methods, then the use of ELO is meaningful and a good solution. However, this is not clear from the explanation in the appendix. I suggest including how these ELO scores are obtained i.e., 1) across how many trials each pair of methods is compared (is it the full 129+83 prompts for each pair?), 2) what is the order in which the pairs are compared. ELO scores highly depend on these factors, especially if there is such a low number of methods in the comparison.
> >
> > **W2**. I appreciate the response, but I don’t think you’ve fully addressed my concerns. I agree that most prompts in LLM applications are benign and using benign prompts from the AlpacaEval dataset is useful and meaningful for assessing helpfulness, as it is rather unlikely for the model to output anything harmful to benign prompts. My concern is about using red-teaming prompts to only assess harmlessness. I don’t think helpfulness and harmlessness are mutually exclusive. When presented with a ‘harmful’ prompt, the model should foremost not output a harmful response, but meanwhile remain helpful, by explaining relevant nuances about the subject matter. Here’s an example to illustrate my point: if a model learns to detect every red-teaming prompt as harmful, then it can simply give a blank response. Under your separate evaluation approach of only assessing harmfulness, this model would be 100% harmless. This means that the best-performing model only needs to detect when it's being red-teamed and remain silent, and in the meantime provide maximally helpful answers for general benign prompts. Given the example responses from your methods in the appendix, this is, of course, not the case, but the principled point stands regarding the chosen decoupled evaluation approach. Hence, I still believe both criteria should be considered.
> >
> > “First, when evaluations of helpfulness and harmfulness are coupled, safe models are likely to be evaluated as helpful. This means that safety-aligned models potentially obtain an unreasonably high evaluation regarding helpfulness. This is based on our observations in early experiments that DPO (S) were valued as more helpful than we humans thought.” All of this seems more like a problem of a lack of a reliable evaluation method, which further justifies my concern raised in W4.
> >
> > **W3 & Q3**. Thank you for mentioning the training time and dataset usage. Let me further elaborate on my concerns regarding this. Although the wall-clock time, which you provided, is definitely a useful indicator, what I am after here is a more rigorous overview i.e., does one stage of training last until a) the model reaches convergence? b) a fixed number of policy updates have been carried out? c) every datapoint has been incorporated into the training? This is currently left up for interpretation. These details are vital for reproducibility and facilitating a fair comparison for future works. For instance, adding training curves in the main paper or appendix would already greatly remedy this issue. In the main paper, all I can find is “We utilize the PKU-SafeRLHF preference dataset with more than 30,000 expert evaluations”. This alone is not very indicative. Appendix G3 solely presents the hyperparameters which was not really what I addressed in my previous inquiry. I can see that each stage was trained for 3 epochs, but there’s no telling what constitutes an epoch. Although, of course, it is great that the source code is accessible, I don’t think it should be expected for readers to go digging in the code for these important details.
> >
> > **W4**. Thanks for including the reference. It does make it more justifiable since it has already been widely used. However, it would require additional analysis to establish LLMs as competent evaluators for safety alignment. For instance, I am curious whether evaluating the same answer 10 times would yield the same helpfulness/safety score from the LLM. I acknowledge that human evaluators are also imperfect in this matter, as they might assign a different score to the same problem under different internal or external circumstances. However, it is unclear how drastic this variance would be for GPT-4. I don’t see any guarantee why GPT-4 wouldn’t potentially flip its preference between two presented answers on every consecutive trial. Or, for instance, whether the LLM would suffer from anchorship bias and grant one answer a very high score, because the other is comparatively much worse, although both answers are objectively very unhelpful. Note that my concern here is not how much SACPO outperforms SafeRLHF or SFT, but that evaluation solely relying on LLMs (especially only a single model and a single trial) is not a reliable metric (at least not yet).
> >
> > As the core of my main concerns (W2 & W4) remain unaddressed, **I am currently unwilling to raise the score**.

---

> > > ### Author Response · Authors · 2024-08-08
> > >
> > > We would like to express our sincere gratitude to Reviewer w7hx who read through our responses.
> > >
> > > **Q1.** Thank you for your helpful suggestions! In our early experiments, we also noticed that ELO scores highly depend on the comparison order and implemented our evaluation protocol to address this. For each ELO score, we conduct 50 evaluation trials. The final score is the average of these trials. To address the inconsistency caused by comparison order, we shuffle the order in each trial. We used all 128 benign prompts for each pair when evaluating helpfulness and all 83 red-teaming prompts for each pair when evaluating harmlessness. Furthermore, to increase the consistency in each trial, we also duplicated these evaluation results 10 times before computing each trial's ELO score. We will include the details on how we obtained the ELO scores in the camera-ready version.
> > >
> > > **W2.** We have conducted additional helpfulness evaluation using red-teaming prompts. We make GPT-4 evaluate each response pair three times and take the mean score.
> > >
> > > | Model | Helpfulness win-rate (red-teaming prompts) | Harmlessness win-rate |
> > > |-------|--------------------------------------------|-----------------------|
> > > | alpaca-sft | 0.50 | 0.50 |
> > > | beaver-v1  | 0.76 | 0.70 |
> > > | DPO(H)    | 0.62 | 0.49 |
> > > | SACPO: DPO(H) -> DPO(S) 0.05  | 0.74 | 0.70 |
> > > | SACPO: DPO(H) -> DPO(S) 0.025 | 0.84 | 0.81 |
> > > | P-SACPO: q=0.75 | 0.84 | 0.77 |
> > >
> > > In this evaluation setting, SACPO and P-SACPO still outperform alpaca-sft and beaver-v1, showing that the models trained by SACPO and P-SACPO can simultaneously provide helpful and harmless responses versus harmful prompts. Another important observation is that the helpfulness score of DPO(H) is comparatively low. We observed that GPT-4 often gives low helpfulness scores for the 'informative but harmful' responses produced by DPO(H). We hope these results can elaborate on the concerns and explain why we use benign prompts for evaluating helpfulness. On the other hand, we agree that such evaluation should be considered, and we will include this evaluation result in the camera-ready version.
> > >
> > > **W3 and Q3.**
> > > Thank you for your constructive feedback! In each stage, we trained each model for a fixed number of epochs, as shown in Appendix G.3. An epoch means one complete pass through the entire training dataset. We also confirmed that our models converged with the chosen epoch numbers.  We will add these important details in the camera-ready version.
> > >
> > > **W4.** We have taken much care of the variance of evaluations by GPT-4. In our paper (Appendix G.6), we have included the experimental results of significance tests by running each evaluation three times. We observed and confirmed that the evaluations by GPT-4 are fairly consistent in our experimental settings.
> > >
> > > We appreciate your valuable suggestions and feedback for improving the quality of our manuscript.

---

> > > > ### Comment · Reviewer_w7hx · 2024-08-09
> > > >
> > > > This is not one of my initial concerns, but there are several issues **and a potential major flaw** with **Additional Experiment 2: SACPO with multiple safety signals** in the global response:
> > > >
> > > > 1. I am willing to accept the definition that harmlessness is considered as safety, but to correlate verbosity with safety is simply not accurate. Note, that I don’t mind that verbosity is chosen as an additional constraint, but that it is interpreted as a safety signal. I would redefine it as a multi-constraint or double-constraint alignment problem.
> > > > 2. To make this experiment fully credible, it is also necessary to show other alignment sequences, not only DPO → (H) → DPO(S) → DPO(V). One of the claims of the paper is that SACPO facilitates any order of alignment and can be combined with any optimization algorithm. Therefore, this should equally apply to the double-constraint scenario.
> > > > 3. The results are presented improperly. If helpfulness and safety are evaluated in a pair-wise comparison, why is verbosity an exception? It would be more coherent to maintain the same evaluation strategy. Additionally, this would allow for a 3D plot for visualization, although I am not expecting this to fit in the single-page additional results PDF.
> > > > 4. **GPT-4 seems to assess helpfulness solely based on the word count.** From the results in Table 1, we can clearly observe that the helpfulness achieved with SACPO starts to closely match that of the base SFT model as the generation lengths become more equal. This further diminishes the credibility of the chosen evaluation metric and begs the question of whether the helpfulness alignment with SACPO actually leads to more helpful answers not just lengthier ones. As the authors themselves stated:
> > > > > Although we observed a slight reduction in helpfulness, this can be related to the issue that GPT-4 evaluation tends to prefer longer answers.
> > > >
> > > > **W2**. Thank you for providing the extended results. This sufficiently addresses my concern. I believe it is important to demonstrate that SACPO can remain helpful while dealing with harmful prompts.
> > > >
> > > > **W3**. I appreciate the clarification. This satisfies my concern.
> > > >
> > > > **W4**. Thank you for pointing out the significance testing experiments. This indeed resolves the concern regarding the evaluation stability. However, in addition to my initial concerns about using a single LLM as the sole evaluator, the problems keep re-emerging, especially the one I addressed above. Moreover, in your response to W2, it was mentioned that
> > > >
> > > > > GPT-4 often gives low helpfulness scores for the 'informative but harmful' responses.
> > > >
> > > > This is not an issue with the dataset nor the strategy of evaluating helpfulness and harmlessness together, but once again that solely relying on an LLM for evaluation is inadequate. The deficient evaluation may jeopardize the results and the conclusions drawn from them. I still don’t think that GPT-4 is a fully quantitative and reliable metric. This substantially decreases the quality of the paper.
> > > >
> > > > **I urge the authors to show that GPT-4 does not evaluate helpfulness solely based on the output length of the model or I may be inclined to lower the score.**

---

> ### Author Response · Authors · 2024-08-11
>
> First of all, we would like to emphasize that **our paper mainly focuses on a problem with a single safety constraint**.
> In relation to multiple safety constraints, **we have shared the same perspective as the initial comments in Strength by Reviewer w7hx.**
>
> > S2. While the authors primarily focus on a single safety function to constrain LM harmfulness and do not empirically analyze scenarios with multiple safety constraints, they provide a theoretical framework that outlines how SACPO can be extended to accommodate multiple safety constraints. This facilitates further extensions of this work.
>
> Please note that Additional Experiment 2 was conducted **not to support the main claims of our paper but as a supplementary effort** to address the queries raised by Reviewer VoMk and Reviewer 4XGF.
>
> > Reviewer VoMk: Can SACPO be generalized to the setting of multiple safety signals? I.e., when using multiple metrics (toxicity, bias, privacy, ...) to measure the harmlessness instead of one single cost.
>
> > Reviewer 4XGF: What are some scenarios that involve multiple constraints? Can the authors provide experiments to illustrate these scenarios?
>
> **Responses on Additional Experiments 2.**
> We will now address the questions and comments regarding Additional Experiment 2. Given the limited time frame to complete the experiment, we focused on verbosity bias as an example. This metric was selected because it can be easily evaluated by simply counting the number of words, unlike metrics such as helpfulness or harmlessness, which require pairwise comparison.
> Our interpretation of the queries from Reviewers VoMk and 4XGF was that they were inquiring whether our SACPO framework can handle multiple safety metrics. Therefore, we decided to prioritize DPO (H) $\rightarrow$ DPO (S) $\rightarrow$ DPO (V) while deprioritizing experiments on alignment orders to ensure that all additional experiments, including Additional Experiments 1 and the one for Reviewer z9np, were completed within the given timeframe.
>
> Regarding the evaluation, while several papers (e.g., Dubois+, Saito+) have recently pointed out the verbosity bias in LLMs, to the best of our knowledge, they do not claim that GPT-4 evaluates helpfulness *solely* on generation length.
>
> - Dubois et al. "Length-controlled alpacaeval: A simple way to debias automatic evaluators." arXiv preprint arXiv:2404.04475 (2024).
> - Saito et al. "Verbosity bias in preference labeling by large language models." arXiv preprint arXiv:2310.10076 (2023).
>
> Based on these existing research findings, we believe that it is more reasonable to regard that GPT-4 does not evaluate helpfulness solely based on generation length.
> Additionally, longer responses may be perceived as more helpful since they can contain more information, and humans often prefer more detailed answers.
>
> Finally, we emphasize that the verbosity bias is not a primary focus of this paper; it serves merely as an example of a further extension of our work.
>
> **W4.** As mentioned during this rebuttal period, the evaluation by GPT-4 is a popular and standard approach, which was especially true at the time of NeurIPS submission. GPT-4 has been shown to be well-aligned with and is a good proxy for human evaluation. For example, in Section 6.4 of the DPO paper (NeurIPS version), they mentioned that “GPT-4 tends to agree with humans about as often as humans agree with each other, suggesting that GPT-4 is a reasonable proxy for human evaluations.” We can also find a similar discussion in Section 4.2.1 of the Safe-RLHF paper, which we used the same evaluation: “When compared to Alpaca-7B, the Beaver-v3 model demonstrated an increase in the Elo score for helpfulness (GPT-4: +244.91, Human: +363.86) and for harmlessness (GPT-4: +268.31, Human: +237.98). Comparatively, the evaluations by GPT-4 and human evaluators are almost consistent.” Based on such previous work on LLM-as-a-Judge or our statistical test, we consider our evaluation sufficiently reliable. That said, as the reviewer mentions, it will be more reliable and unbiased to evaluate using multiple LLMs.
>
> Thank you for your valuable comments. Finally, we would be grateful if you could look again at not only the weaknesses of our paper but also the strengths or concerns we have addressed during the rebuttal period.

---

> ### Comment · Reviewer_w7hx · 2024-08-13
>
> > Please note that Additional Experiment 2 was conducted not to support the main claims of our paper but as a supplementary effort to address the queries raised by Reviewer VoMk and Reviewer 4XGF.
>
> Thank you for highlighting this. I do not consider it as a weakness.
>
> > ...they do not claim that GPT-4 evaluates helpfulness solely on generation length...
>
> Although not explicitly claimed by prior works, my concern still remains that the response length may have a strong impact on the evaluation of helpfulness.
>
> > longer responses may be perceived as more helpful since they can contain more information
>
> This is definitely true, however it is not guaranteed that a long response can also simply reiterate the core helpful parts, and is thus perceived by GPT-4 as more helpful.
>
> **W4**. Thank you for bringing these references to my attention. They do justify the evaluation strategy more. Although GPT-4: +244.91 - Human: +363.86 seems like quite a substantial gap.
>
> Considering the other reviews, the authors' responses to them, and all of my concerns almost fully addressed, I have decided to **increase the score**.

---

> > ### Author Response · Authors · 2024-08-13
> >
> > We would like to express our sincere gratitude to Reviewer w7hx for the insightful comments and constructive suggestions. Based on the valuable feedback received during the initial review and discussion period, we will complete the camera-ready paper, ensuring that the information is accurately conveyed to our readers. Your thoughtful remarks and feedback have greatly contributed to enhancing the quality of our manuscript, and we are truly appreciative of your efforts.

---

### Official Review · Reviewer_z9np · 2024-07-13

**Soundness:** 3
**Presentation:** 3
**Contribution:** 3
**Rating:** 6
**Confidence:** 4

**Summary:**

From the perspective of safe reinforcement learning, the author formulates human value alignment as an optimization problem of the LM policy to maximize reward under a safety constraint, and then proposes an algorithm, Stepwise Alignment for Constrained Policy Optimization (SACPO).

**Strengths:**

The author introduces SACPO, an algorithm that effectively enhances the safety of LLMs. The theoretical derivations are solid, and the algorithm's effectiveness is demonstrated through extensive experimental settings.

**Weaknesses:**

1. Although the author included some descriptions of related work in the Preliminaries section, I did not find a dedicated Related Work section in the main paper. This omission significantly impairs readability. It is unclear why the author chose to exclude this section from the main paper. Given that the paper explores the safe alignment of llms from the perspective of safe reinforcement learning, two highly relevant areas of related work would be Safe Reinforcement Learning and Safety Alignment of LLMs.
2. Considering the point 1, I am curious about the relationship between the proposed SACPO algorithm and traditional safe reinforcement learning algorithms. I noticed that the author describes SACPO's two phases in Algorithm 1 as reward alignment and safety alignment. In traditional safe reinforcement learning, such as in the Constrained Update Projection Approach to Safe Policy Optimization, the algorithm's update logic is similarly divided into Reward Improvement and Projection (safety satisfaction). What are the connections between the proposed algorithm and traditional reinforcement learning algorithms, or the challenges and difficulties in extending traditional reinforcement learning algorithms to the LLMs setting? From a reviewer’s perspective, these points are worth including in the main text.
3. I am also working on LLMs Safety Alignment and appreciate the motivation behind this work. I observed that in the experimental section, the author aligns SACPO with DPO(H) and DPO(S), where the latter two are trained separately using the Helpful and Harmless dimensions from Beavertails. Since Beavertails' preference annotations are decoupled—helpfulness is annotated without considering safety—DPO models trained separately are naturally deficient in the corresponding dimensions. I would be interested to see a comparison between DPO models trained with a trade-off between helpfulness and harmlessness and SACPO. For example, how do they perform on datasets like PKU-SafeRLHF and PKU-SafeRLHF-single-dimension?
>https://huggingface.co/datasets/PKU-Alignment/PKU-SafeRLHF
https://huggingface.co/datasets/PKU-Alignment/PKU-SafeRLHF-single-dimension

**Questions:**

see above.

**Limitations:**

see above.

---

> ### Author Rebuttal · Authors · 2024-08-06
>
> We deeply appreciate the reviewer’s encouraging comments. The thoughtful comments of Reviewer z9np are valuable and help us improve the manuscript's quality. We will answer comments in Weaknesses.
>
> **Weakness 1.**
> We appreciate the reviewer's feedback. Given the amount of content and the strict page limit, we decided to omit the related work section and review particularly relevant existing studies (e.g., RLHF, DPO, Safe RLHF) as preliminaries for understanding our SACPO. Based on the reviewer's comments, we will add more discussion on related work in the camera-ready version.
>
> **Weakness 2.**
> Thank you for your constructive feedback. As the reviewer mentions, safe reinforcement learning (RL) is a highly relevant topic to our paper. Similar to safe RL algorithms (based on constrained criteria), our SACPO also tries to optimize a policy to maximize a reward under safety constraints. Also, our theoretical analyses are largely based on the literature on safe RL or constrained Markov decision processes (CMDPs). On the other hand, differences and challenges exist specific to LLM alignment settings. As for problem formulation, we consider a setting of a contextual bandit with safety constraints, which is a special case of general safe RL. This is a simplified setting and we do not have to deal with challenges specific to RL (e.g., state transitions). However, the scale of the experiments (e.g., size of neural networks, amount of dataset) is significantly larger compared to solving typical safe RL benchmark tasks such as Safety-Gym (Ray et al., 2019).
>
> - Ray, Alex, Joshua Achiam, and Dario Amodei. "Benchmarking safe exploration in deep reinforcement learning." arXiv preprint arXiv:1910.01708 7.1 (2019): 2.
>
> Having this difficulty in mind is critical in developing LLM alignment algorithms, which leads to our proposal of P-SACPO based on model merging as an example. We agree with the reviewer that the RL community is now interested in the intersection between RL and LLM, and this paper should be written in a way friendly to the (safe) RL community. Therefore, we will discuss the connections and differences between safe RL and our paper in the camera-ready version while referring to the following paper the reviewer mentions.
>
> - Yang, Long, et al. "Constrained update projection approach to safe policy optimization." Advances in Neural Information Processing Systems 35 (2022): 9111-9124.
>
> **Weakness 3.**
> Thank you for your insightful feedback! We have conducted an additional experiment based on your comments. In this experiment, we used the single-dimension version of the PKU-SafeRLHF-30K dataset and applied DPO to the same SFT model. The following table summarizes the win rates against the SFT model. Note that SACPO means DPO (H) $\rightarrow$ DPO (S) with $\lambda/\beta = 0.025$.
>
> |  | Win-rate (Helpfulness) | Win-rate (Safety) |
> | ---- | ---- | ---- |
> | SACPO | 0.685 | 0.805 |
> | DPO (single-dimension) | 0.640 | 0.741 |
>
> DPO (single-dimension) requires using the same $\beta$ for reward and safety. However, SACPO allows us to use different KL penalty coefficients (i.e., $\beta$ for reward and $\beta/\lambda$ for safety).

---

### Official Review · Reviewer_VoMk · 2024-07-14

**Soundness:** 3
**Presentation:** 3
**Contribution:** 3
**Rating:** 6
**Confidence:** 4

**Summary:**

This paper proposes a new alignment algorithm SACPO to improve the both helpfulness and safety (harmlessness) of language model. SACPO separates the two objectives into two alignment steps. The second step of optimization is equivalent to be an optimization with the policy from first step as the reference policy. Meanwhile, each step can be implemented with reward-free alignment methods (e.g., DPO, KTO). The authors compare the proposed method with SFT base model and show it can achieve higher win rate in terms of harmlessness and helpfulness.

**Strengths:**

- Overall, the paper is clearly written and easy to follow.
- The two steps optimization of SACPO can be achieved by reward-free alignment algorithm, which reduces the requirement of computation source and dataset (e.g., it also works for dataset which is not constructed by preference data).
- The empirical results show that proposed method exceeds the baseline on helpfulness and harmlessness.

**Weaknesses:**

- Although the authors use P-SACPO as a remedy, it is still questionable that the proposed method cannot get the correct $\lambda^*$. For example, how to step a conservative starting point $\bar\lambda$, how to set the linear interpolation coefficient $q$.
- As a general alignment algorithm, the authors should test the performance on other datasets or different models.

**Questions:**

- Can SACPO be generalized to the setting of multiple safety signals? I.e., when using multiple metrics (toxicity, bias, privacy, ...) to measure the harmlessness instead of one single cost.
- What are the differences of three subfigures in fig.2? Are they using the same measurements (i.e., are the points in different subfigures comparable)?
- Could you compare the win rate of SACPO and baseline against stronger model (e.g., compare it to gpt 3.5 and gpt 4.0 as AlpacaEval does)?
- As Remark 2 indicates, the order of two-step alignment will not influence the final policy in theory. However, in fig.2(b), the performances of DPO(H->S) and DPO(S->H) for $\beta/\lambda=0.1$ are very different. Do you have any explanation for that?

**Limitations:**

The authors have mentioned several limitations of this work in Sec.8. One another limitation is that the proposed method only works for the KL divergence regularizer ($D_{KL}[\pi(\cdot|x)\|\pi_{ref}(\cdot|x)]$) as discussed in Remark 1.

---

> ### Author Rebuttal · Authors · 2024-08-06
>
> We deeply appreciate the reviewer’s encouraging comments. The thoughtful comments of Reviewer VoMk are valuable and help us improve the manuscript's quality.
> We will answer the Questions first and then address the comments in Weakness.
>
> **Questions**
> > **Q1.** Can SACPO be generalized to the setting of multiple safety signals? I.e., when using multiple metrics (toxicity, bias, privacy, ...) to measure the harmlessness instead of one single cost.
>
> Thank you for the important question! Theoretically, SACPO can generalize to the setting of multiple safety signals as discussed in Appendix B in a theoretically justified manner. Empirically, our paper did not validate whether this theoretical finding is true in practice. Therefore, we have conducted an additional experiment with multiple safety signals. Specifically, we aligned LLMs so that the verbosity bias is mitigated after enhancing helpfulness and harmlessness. **For more details, we would like to ask the reviewer to look at the global response.**
>
> >**Q2.** What are the differences of the three subfigures in fig.2? Are they using the same measurements (i.e., are the points in
> different subfigures comparable)?
>
> We first tried to create a single figure, but the resulting figure was messy due to many data points. To improve the visibility of the experimental results and clarify the difference between each method, we separated the single figure into three subfigures. The three figures use the same measurements for several items such as the red points (i.e., DPO (H) $\rightarrow$ DPO (S)), black points (i.e., SFT model), or black crosses (i.e., models by Safe RLHF).
>
> >**Q3.** Could you compare the win rate of SACPO and baseline against stronger model (e.g., compare it to gpt 3.5 and gpt 4.0
> as AlpacaEval does)?
>
> Our primary objective is to propose a new safety-alignment algorithm rather than releasing models with state-of-the-art performances. Thus, we take much care to ensure a fair comparison between our proposed SACPO and baseline methods (e.g., Safe RLHF) while using the same base SFT model and training dataset. Our experiments used LLMs with 7 billion parameters, and our resulting models are much smaller than GPT 3.5 or GPT 4. For a fair comparison with such GPTs, we will need to apply our SACPO to LLMs with a comparable size of parameters, but it is impossible given our budget or computational resources. As discussed in Section 7, we admit that it is an important future direction to investigate whether SACPO works well for state-of-the-art models with many more parameters.
>
> >**Q4.** As Remark 2 indicates, the order of two-step alignment will not influence the final policy in theory. However, in g.2(b),
> the performances of DPO(H$\rightarrow$S) and DPO(S$\rightarrow$H) for $\beta/\lambda$ are very different. Do you have any explanation for that?
>
> Thank you for the great question! We have also wondered why the performances of DPO (H) $\rightarrow$ DPO (S) and DPO (S) $\rightarrow$ DPO (H) for the same $\beta/\lambda$ are very different. We guess that the poor representation ability of the LLMs or optimization error regarding DPO would lead to such a phenomenon, but we do not have a clear explanation. It is an interesting direction to analyze such a gap between theory and practice, which we will leave to future work.
>
> **Weaknesses**
> >**W1.** Although the authors use P-SACPO as a remedy, it is still questionable that the proposed method cannot get the correct $\lambda^\star$. For example, how to step a conservative starting point $\bar{\lambda}$, how to set the linear interpolation coefficient $q$.
>
> We empirically observed that $\bar{\lambda} = 10$ (i.e., $\beta/\bar{\lambda} = 0.01$ in the case of $\beta = 0.1$) is highly likely to result in sufficiently conservative language models.
> Though we do not feel difficulty in choosing $\bar{\lambda}$, there might be cases where humans need to tune $\bar{\lambda}$ depending on the base model or dataset.
> However, typical implementations for (safety-agnostic) DPO use a parameter $\beta$ around $0.1$, which is common knowledge in the community. While the community also accumulates knowledge regarding safety alignment, we consider that it will be easier to obtain such a conservative model. Regarding the linear interpolation coefficient, humans can choose $q$ to maximize reward (i.e., helpfulness) under safety constraints. Our P-SACPO does not necessitate additional training or fine-tuning; hence, it is not costly to try a wide range of parameters of $q$ while simply merging two models.
>
> >**W2.** As a general alignment algorithm, the authors should test the performance on other datasets or different models.
>
> Thank you for your insightful comments! We have conducted additional experiments using a different base SFT model and dataset. Regarding this, we conducted two experiments: one used LLaMA2 and hh-rlhf datasets, and the other used Pythia and the PKU-SafeRLHF-30K dataset. **For more details, we would like the reviewer to see the global responses** and we hope our new results resolve the reviewer's concerns.
>
> **Limitation**
> Thank you for pointing out additional limitations! We will add the limitation suggested by the reviewer in the camera-ready version.

---

> > ### Comment · Reviewer_VoMk · 2024-08-09
> >
> > Thanks for your response. My concerns have been addressed and I will keep my score.

---

> > > ### Author Response · Authors · 2024-08-11
> > >
> > > We are glad to hear that Reviewer VoMk's concerns have been addressed. We would like to express our sincere gratitude to the reviewer who read through our responses and other reviews.

---

### Official Review · Reviewer_KBZp · 2024-07-15

**Soundness:** 3
**Presentation:** 3
**Contribution:** 3
**Rating:** 7
**Confidence:** 4

**Summary:**

The paper addresses the challenge of fine-tuning a language model (LM) policy to maximize reward while adhering to safety constraints. Building on the concept of Safe RLHF, which introduces a constrained safe RL paradigm for aligning LLMs, the authors propose a novel approach: Stepwise Alignment for Constrained Policy Optimization (SACPO). Unlike the traditional method that simultaneously balances reward and safety optimization, SACPO adopts a stepwise approach, first aligning the LLM for reward and then for safety, or vice versa. They also present a practical variant called P-SACPO, which leverages model merging techniques. Empirical results demonstrate the superiority of SACPO over baseline methods such as Supervised fine tuning (SFT) and Safe RLHF.

**Strengths:**

Merites of the proposed method (SACPO): (1) simple, stable, and computationally efficient compared to Safe RLHF; (2) compatible with different alignment algorithms (DPO, KTO, and IPO) and datasets; (3) it has solid theoretical grounding.

**Weaknesses:**

Experiments on additional datasets could strengthen the paper's findings.

**Questions:**

In the experiments, you have tested only four combinations (DPO (H) → DPO (S), DPO (H) → KTO (S), KTO (H) → DPO (S), and DPO (S) → DPO (H)). Why other potential variants, such as KTO (H) → KTO (S), were not considered?

**Limitations:**

In the conclusion, the authors acknowledge the limitations of their work and the potential societal impact.

---

> ### Author Rebuttal · Authors · 2024-08-06
>
> We deeply appreciate the reviewer KBZp for helpful and thoughtful comments and questions.
> We will first answer the Question and then address the comments in Weakness.
>
> **Questions**
>
> >In the experiments, you have tested only four combinations (DPO (H) $\rightarrow$ DPO (S), DPO (H) $\rightarrow$ KTO (S), KTO (H) $\rightarrow$ DPO (S), and DPO (S) $\rightarrow$ DPO (H)). Why other potential variants, such as KTO (H) $\rightarrow$ KTO (S), were not considered?
>
> In our experiments, KTO (H) performed worse than DPO (H) in terms of alignment to enhance helpfulness (i.e., reward). Even worse, DPO (H) $\rightarrow$ KTO (S) performed significantly worse than DPO (H) $\rightarrow$ DPO (S). Therefore, we did not try KTO (H) $\rightarrow$ KTO (S) because we thought it was unlikely to work well.
>
> **Weakness**
>
> >Experiments on additional datasets could strengthen the paper's findings.
>
> Thank you for your constructive feedback! We agree with the reviewer that experiments on additional datasets could strengthen the paper's findings. Therefore, we conducted additional experiments using a different base SFT model and dataset. **New experimental settings and results are presented in the global response.** Especially, we conducted an experiment using the hh-rlhf dataset (which is different from the PKU-SafeRLHF-30K dataset we initially used), and we hope the new results will resolve the reviewer's concern.

---

> > ### Comment · Reviewer_KBZp · 2024-08-11
> >
> > Thank you for your response and additional experiments! I will increase my score.

---

> > > ### Author Response · Authors · 2024-08-11
> > >
> > > We are delighted to hear that Reviewer KBZp's concerns have been addressed! We extend our heartfelt gratitude to the reviewer for taking the time to consider our responses carefully!

---

### Author Rebuttal · Authors · 2024-08-06

Dear reviewers and AC,

We deeply thank all the reviewers for their insightful comments and constructive suggestions.

- We have conducted new experiments based on the reviewers' comments. Additional experimental results are provided in a one-page PDF containing new figures attached in this "global" response.
We will present the details of the additional experiments below.

- We have also provided our detailed response to each reviewer with a separate response.

We hope our replies have addressed all the questions and concerns of the reviewers. We are willing to answer any of the reviewers' concerns about our work.

Best regards,

Authors.


---

### **Additional Experiment 1: SACPO with different base SFT models and datasets**

In this experiment, we additionally tried two settings to assess the performance of SACPO with diverse base SFT models and datasets.

**LLaMA2 (7B) model + Anthropic/hh-rlhf.** In the first setting, we employed the LLaMA2 (7B) model and the Anthropic/hh-rlhf preference dataset. Note that the Anthropic/hh-rlhf dataset is constructed by several subsets, including harmless-base, helpful-base, helpful-online, helpful-rejection-sampled, and red-team-attempts. First, we conducted supervised training using randomly selected 100K samples of the whole hh-rlhf dataset. Then, using the 'helpful-base' subset, we conducted helpfulness alignment with DPO on this SFT model. For the safety alignment, we applied DPO for the helpfulness-aligned model using the 'harmless-base' subset. Similar to our main experiment, we used $\beta=0.1$ in the helpfulness alignment phase. In the safety alignment phase, we employed $\beta \in \{0.1, 0.05\}$. The following tables show the parameters different from the experimental settings in the main paper:

|Phase | lr | epochs |
|---|---|---|
|SFT|5e-7|1|
|Helpfulness alignment|5e-6|2|
|safety alignment|5e-6|2|

Figure 1(a) in the one-page PDF shows the helpfulness and safety win rate against the SFT model. We can see that DPO(H) improved helpfulness at the first step but significantly reduced the model's harmlessness. After aligning for safety in the second step, we obtained a large improvement in harmlessness with a slight decrease in helpfulness.

**Pythia-6.9b + PKU-SafeRLHF-30K.** Second, we employed the EleutherAI/Pythia-6.9b model and the PKU-SafeRLHF-30K dataset.
The EleutherAI/Pythia-6.9b model is based on a different architecture compared to the alpaca-7b-reproduced model used in the main experiment, and it is trained on a different dataset.
First, we conducted helpfulness alignment with DPO on the EleutherAI/Pythia-6.9b model and then conducted safety alignment.
The following tables show the parameters different from the experimental settings in the main paper.

|Phase | beta | lr | epochs |
|---|---|---|---|
|Helpfulness alignment|0.05|1e-6|2|
|Safety alignment|0.01|2e-5|2|

Figure 1(b) of the one-page PDF shows the helpfulness and safety win rate against the SFT model.
We can see that DPO(H) improved helpfulness at the first step but significantly reduced the model's harmlessness.
After aligning for safety in the second step, we obtained a slight increase in helpfulness and a significant improvement in harmlessness.

In conclusion, SACPO could obtain a model that performs better than the SFT in terms of helpfulness and harmlessness.
Therefore, we can say that **SACPO performs well as a general alignment algorithm, as evidenced by the initial experimental results and additional ones.**

---

### **Additional Experiment 2: SACPO with multiple safety signals**

In this experiment, we aim to align an LLM to reduce verbosity bias while enhancing the two metrics we have already considered (i.e., helpfulness and harmlessness). To achieve this, we further align a model that has already been aligned for helpfulness and harmlessness. In particular, we employ the model DPO\,(H)\,$\rightarrow$\,DPO\,(S) where $\beta=0.1$ and $\beta/\lambda=0.05$.
First, we create a preference dataset that is suitable for reducing verbosity bias while maintaining its helpfulness and harmlessness. We utilize the same PKU-SafeRLHF preference dataset used in other experiments and set the shorter answer as the more preferred one. The following table shows the helpfulness and harmlessness of shorter responses in the PKU-SafeRLHF dataset. We can see that shorter responses have a higher chance of being safer and less helpful. Thus, using this dataset would significantly reduce the helpfulness of the model.

|                  | shorter is more helpful | shorter is less helpful |
| ---|---|---|
| shorter is safer | 2627 | 11377 |
| shorter is less safe | 3187 | 9683 |

To avoid this side-effect, we sample the dataset to obtain a helpful-harmlessness balanced preference dataset. The data distribution of the balanced dataset is as follows:

|                  | shorter is more helpful | shorter is less helpful |
| ---|---|---|
| shorter is safer | 2627 | 3187 |
| shorter is less safe | 3187 | 2627 |

Finally, we apply DPO using this dataset with $\beta \in \{1, 10\}$, $lr=2e-5$ for 3 epochs.

The helpfulness and safety win rates against the SFT model of the aligned model are shown in Figure 2 and Table 1 of the one-page PDF. We also show the generation length of these models in Table 1. **We see that SACPO can successfully balance the three metrics, and the debiased model can produce shorter answers while remaining helpful and harmless.** Although we observed a slight reduction in helpfulness, this can be related to the issue that GPT-4 evaluation tends to prefer longer answers, which has been pointed out in the literature. We will leave it to future work to investigate whether SACPO can handle many more safety signals.

---

### Decision · Program_Chairs · 2024-09-25

**Decision:**

Accept (poster)

**Comment:**

This paper introduces a new algorithm to tackle the alignment of language models. In particular, it considers a constrained setting where the model maximizes the reward while keeping the safety constraint violations bounded.

The proposed algorithm SACPO adopts a novel stepwise approach that first aligns the language model with reward and then with safety, which allows the use of simpler methods for optimization compared with prior work.

The reviewers are positive about the paper, mentioning that it is well written and the contribution is well positioned and properly supported theoretically and empirically. Furthermore, the rebuttal extended the evaluation with new models and datasets and included an experiment with more constraints, showing that the algorithm performs well under different settings and can balance multiple metrics.

In general, I believe this paper makes a clear contribution to the alignment of language models. Considering the positive support of the reviewers, I recommend accepting it.